

# Estrogen receptor 1 gene polymorphisms (PvuII and XbaI) are associated with type 2 diabetes in Palestinian women

Suheir Ereqat[1], Stéphane Cauchi[2,3,4,5], Khaled Eweidat[1], Muawiyah Elqadi[1] and Abedelmajeed Nasereddin[6]

[1] Biochemistry and Molecular Biology Department-Faculty of Medicine, Al-Quds University, East Jerusalem, Palestine
[2] CNRS, UMR8204, Lille, France
[3] INSERM, U1019, Lille, France
[4] Université de Lille, Lille, France
[5] Institut Pasteur de Lille, Centre d'Infection et d'Immunité de Lille, Lille, France
[6] Al-Quds Public Health Society, East Jerusalem, Palestine

## ABSTRACT

**Background:** Type 2 diabetes mellitus (T2DM) is a multifactorial disease where both genetic and environmental factors contribute to its pathogenesis. The PvuII and XbaI polymorphisms of the estrogen receptor 1 (ESR1) gene have been variably associated with T2DM in several populations. This association has not been studied in the Palestinian population. Therefore, the aim of this study was to investigate the association between the PvuII and XbaI variants in the *ESR1* and T2DM and its related metabolic traits among Palestinian women.

**Methods:** This case–control study included 102 T2DM and 112 controls in which PvuII and XbaI variants of the *ESR1* gene were genotyped using amplicon based next generation sequencing (NGS).

**Results:** Allele frequencies of both PvuII and XbaI variants were not significantly different between patients and control subjects ($P > 0.05$). In logestic regression analysis adjusted for age and BMI, the *ESR1* PvuII variant was associated with risk of T2DM in three genotypic models ($P < 0.025$) but the strongest association was observed under over-dominant model (TT+CC vs. TC) (OR = 2.32, CI [1.18–4.55] adjusted $P = 0.013$). A similar but non-significant trend was also observed for the *ESR1* XbaI variant under the over-dominant model (AA+GG vs. AG) (OR = 2.03, CI [1.05–3.95]; adjusted $P = 0.035$). The frequencies of the four haplotypes (TA, CG, CA, TG) were not significantly different in the T2DM patients compared with control group ($P > 0.025$). Among diabetic group, an inverse trend with risk of cardio vascular diseases was shown in carriers of CG haplotype compared to those with TA haplotype (OR = 0.28, CI [0.09–0.90]; adjusted $P = 0.035$). Further, stratified analyses based on *ESR1* PvuII and XbaI genotypes revealed no evidence for association with lipid levels (TC, TG, HDL, LDL).

**Conclusions:** This is the first Palestinian study to conclude that *ESR1* PuvII and XbaI variants may contribute to diabetes susceptibility in Palestinian women. Identification of genetic risk markers can be used in defining high risk subjects and in prevention trials.

Corresponding author
Suheir Ereqat,
sereqat@staff.alquds.edu

# INTRODUCTION

Type 2 diabetes mellitus (T2DM) is a multifactorial disease that caused by a complex combination of genetic and environmental factors. Identification of genetic polymorphisms associated with diabetes may lead to prediction of disease development and prevention of its vascular complications (*Slominski et al., 2018*). Recently, several reports have revealed the role of estradiol in regulating energy metabolism (*Hevener et al., 2018*). It was evident that estrogens increase hepatic insulin sensitivity, stimulate insulin synthesis in islets of Langerhans, prevent β-cell apoptosis and improve insulin action in skeletal muscles (*Meyer et al., 2011*).

Estrogen exerts its physiological functions through the estrogen receptors (ESR1, ESR2, and a G-protein coupled cell surface receptor) and might prevent menopause syndrome, cardio vascular diseases (CVD) and diabetes (*Gupte, Pownall & Hamilton, 2015*). Estrogen receptor 1 (ESR1) is broadly expressed in adipose tissue, skeletal muscle, liver, and immune cells. It is a ligand-activated transcription factor that regulates a large number of genes in diverse target tissues (*Hevener et al., 2018*). Animal studies reported that male and female *ESR1* gene knockout mice developed features of the metabolic syndrome (MetS) including obesity caused by impaired fatty acid oxidation, glucose intolerance, and impaired insulin sensitivity, thus revealed the critical role of *ESR1* in metabolic homeostasis (*Heine et al., 2000*; *Ribas et al., 2010*).

The *ESR1* gene, which encompasses 140 kb of DNA, is found on chromosomes 6q25.1. It is a highly polymorphic gene containing more than 1,600 single nucleotide polymorphisms (SNPs). Two SNPs in *ESR1*, both located in the first intron, PvuII (rs2234693) and XbaI (rs9340799), are the most extensively investigated variants and reported to be associated with T2DM (*Huang et al., 2006*; *Mohammadi et al., 2013*), MetS (*Zhao et al., 2018*), and other diseases (*Onland-Moret et al., 2005*; *Silva et al., 2010*; *Weng et al., 2015*). These polymorphisms might interfere with the estrogen effect by altering *ESR1* gene expression via altering the binding of its own transcription factors (*Gomes-Rochette et al., 2017*). Diabetes and obesity had reached an alarming rate among Palestinians especially among refugees (*El Kishawi et al., 2014*; *Shahin, Kapur & Seita, 2015*). In 2015, the Palestinian Ministry of Health reported that cardiovascular diseases were the leading cause of death and diabetes mellitus was the fourth cause of death among Palestinians (*Palestinian Ministry of Health, 2015*). A recent study conducted on three refugee camps in the West Bank, revealed that the overall prevalence of obesity and overweight was 63.1% (for obesity: 42% in women and 29.2% in men; overweight 25.8% and 28.9% in women and men; respectively). The prevalence of MetS among obese and overweight adults was 69.4% with no gender-based differences (*Damiri et al., 2018*). Obese and overweight individuals are at high risk of both cardiovascular diseases and T2DM (*Alberti & Zimmet, 1998*). Estrogen deficiency is an important obesity-triggering factor which enhances metabolic dysfunction and thus predisposing to T2DM and cardiovascular diseases among menopausal women (*Lizcano & Guzman, 2014*). A meta-analysis study

investigating the new-onset of T2DM in post-menopausal women following estrogen replacement therapy revealed a 30% lower relative risk [RR 0.7 (CI [0.6–0.9])] of diabetes compared with placebo (*Salpeter et al., 2006*). However, the use of estrogen to prevent chronic diseases is still challenging and controversial (*Bolton, 2016*).

As the *ESR1* gene is a potential candidate gene for susceptibility to T2DM, we hypothesized that the *ESR1* variants (PvuII and XbaI) might be associated with T2DM, diabetic complications and related metabolic traits in Palestinian diabetic women.

## MATERIALS AND METHODS

### Study design and participants

This case control study includes 214 Palestinian women (102 with T2DM and 112 non-diabetic controls). All participants were unrelated, aged >40 years and selected within the period of 2016–2017 from the United Nations Relief and Works Agency for Palestine Refugees clinics (Hebron and Ramallah, Palestine). Diagnosis of T2DM was based on World Health Organization criteria (fasting plasma glucose $\geq$ 126 mg/dl and/or currently on treatment for diabetes) (*Kumar et al., 2016*). The diagnosis of T2DM was confirmed based on patients' medical records reporting initial diagnosis of diabetes mellitus after age of 40, receiving oral hypoglycemic agents without insulin for at least 1 year after initial diagnosis, and currently on treatment for diabetes. Patients with probable type 1 diabetes who received continuous insulin therapy since diagnosis were excluded. The anthropometric measurements were collected from their medical records using a standard questionnaire that included age, age at diagnosis, sex, family history, diabetic complications, and medication for diabetes. BMI was calculated as kilograms divided by the square of height in meters. Blood pressure was measured in sitting position, on the left arm, after a 5-min rest by a health clinic worker, with a mercury sphygmomanometer. Blood samples (five ml) after a 12-h minimum fast was collected in EDTA tubes for biochemical tests as described in our previous work (*Sabarneh et al., 2018*). Plasma glucose, cholesterol, HDL cholesterol, and triglyceride were determined using standard methods of commercial kits (Human, Wiesbaden, Germany). Low-density lipoprotein (LDL) cholesterol was calculated based on the Friedewald formula. The control group was selected from individuals who came to the same clinic for an annual heath check-up, they were eligible to be included if they had no prior diagnosis for T2DM, no family history in first-degree relatives, their fasting glucose < 126 mg/dl and BMI > 25. All the participants provided written informed consent to participate in the study, the study protocol was approved by Al-Quds University Research Ethics Committee (Rf no. 2/SRC/4).

### Amplicon based next generation genotyping and bioinformatics

Genomic DNA was extracted from whole blood (300 μl) using genomic QIAamp DNA purification kit according to the manufacturer instructions (Qiagen, Hilden, Germany). All DNA samples were genotyped for the T/C, rs2234693, and A/G, rs9340799, also known as PvuII and XbaI polymorphisms, respectively using amplicon based next generation sequencing (NGS). Briefly, two primers (forward and reverse) were used to target the two SNPs as previously described (*Motawi et al., 2015*). Both primers were

modified with over hanged Illumina adaptor sequences at the 5′ ends (italic bolded, Table S1) to target a partial sequence of 119 bp in length with a final product of 186 bp using conventional thermocycler.

The PCR product was visualized and captured on a 1.5% agarose gel, cleaned by Agencourt AMPure XP system (X1, A63881; Beckman Coulter Genomics, Indianapolis, IN, USA) and eluted in 25 μl elution buffer. All purified products were subjected to a second round of amplification to assign unique index sequences (barcode) for each sample using Nextera XT Index Kit (Illumina, San Diego, CA, USA). Five μl from each barcoded sample were pooled together, mixed and spin down. Then, 100 μl of the pooled product was cleaned by Agencourt AMPure XP system (X1) (A63881; Beckman Coulter Genomics), and eluted in 50 μl elution buffer. Library purity and quantity were evaluated by 4200 TapeStation System (Agilent Technologies, Inc., Santa Clara, CA, USA) using D1000 ScreenTape kit (Agilent Technologies, Inc., Santa Clara, CA, USA) and by Qubit® Fluorometer (Invitrogen, Carlsbad, CA, USA) using Qubit dsDNA high-sensitivity assay (Invitrogen, Carlsbad, CA, USA). Concentration of four nM was prepared. 20 K reads for each sample was targeted. Samples were deep sequenced on NextSeq 500/550 machine using the 150-cycle Mid Output Kit (Illumina, San Diego, CA, USA).

The obtained DNA sequences were uploaded on the Galaxy program (https://usegalaxy.org/). Workflow of filtration included Illumina adaptor trim, quality selection of $Q > 20$ with minimal read length of 100 bp. Four virtual probe sequences were used to identify the PvuII and XbaI variants (Table S1). The genotypes were determined based on the calculated ratio between the read counts for wild type and mutant alleles, for both SNPs (PvuII and XbaI) in each individual sample.

## Statistical analysis

Statistical analysis was performed with the SPSS package, version 19.0 (SPSS, Inc., Chicago, IL, USA). All tests were two-tailed and $P < 0.05$ was considered significant. The mean values and standard errors were reported in tables. Genotype and allele frequencies in T2DM and control subjects were tested by multivariable logistic regression analysis using five genetic models: co-dominant, dominant, over-dominant, recessive and additive with adjustment for age and BMI using R statistics (V 3.5.1; SNPassoc package) (*Gonzalez et al., 2007*). The genotype frequencies were tested for Hardy–Weinberg equilibrium by calculating a $\chi^2$ statistic and corresponding $P$-value as previously described by *Mayo (2008)*. The risk to T2DM was estimated by computing odds ratios (ORs) and 95% confidence intervals (CIs). Linkage disequilibrium (LD), coefficient (D′) for haplotypes and their frequencies were performed using R statistics (V 3.5.1; SNPassoc package) (*Gonzalez et al., 2007*). Multivariate linear regression models taking into account age and BMI were performed to assess associations between SNPs and T2DM complications.

All quantitative parameters were normalized before analysis using the "bestNormalize" R package (https://github.com/petersonR/bestNormalize) (*Peterson, 2017*). To minimize the chance of obtaining type I error, Bonferroni's adjustment was applied. $P < 0.025$ was adopted as the significant threshold unless otherwise specified. Quanto 1.2.4 software

**Table 1 Demographic and biochemical characteristics of study subjects.**

|  | Control (*n* = 112) | T2DM (*n* = 102) | *P*-value |
|---|---|---|---|
| Age at sampling (years) | 48.9 (0.68) | 59.9 (0.95) | **0.0001** |
| Age at diagnosis (years) | NA | 49.6 (0.83) |  |
| BMI (Kg/m$^2$) | 32.5 (0.52) | 34 (0.6) | 0.054 |
| SBP (mmHg) | 122 (0.93) | 136.2 (1.6) | **0.0001** |
| DBP (mmHg) | 77.9 (0.85) | 78.9 (1.02) | 0.4 |
| FBS (mg/dl) | 90.1 (0.97) | 161.8 (4.9) | **0.0001** |
| TC (mg/dl) | 182 (3.4) | 192 (4.02) | **0.03** |

Notes:
BMI, body mass index; SBP, systolic blood pressure; DBP, diastolic blood pressure; FBS, fasting blood sugar.
Data are presented as mean (standard error); $P < 0.05$ was considered significant (in bold). NA, Not applicable.

was used to estimate the statistical power (http://biostats.usc.edu/Quanto.html). The power of any test of statistical significance was <50%.

# RESULTS

## Biochemical and genetic analysis

Anthropometric and biochemical characteristics of study individuals are presented in Table 1. In comparison with the control group, all clinical parameters showed statistically significant increase in T2DM patients ($P < 0.05$). However, no significant difference in diastolic blood pressure was observed among the two group ($P = 0.4$). The T2DM subjects were significantly older ($P = 0.0001$) but the control group were at the same mean age at T2DM diagnosis in cases ($P > 0.05$) (Table 1). The cases had higher mean BMI than control subjects but was at borderline significance ($P = 0.054$). Among T2DM patients, 10.8, 8.8, 7.8, and 5.9% had CVD, nephropathy, diabetic foot, and retinopathy, respectively. The genotyping distribution for both variants was consistent with Hardy Weinberg equilibrium in cases and control groups and in all subjects ($P > 0.05$). Our results showed that the heterozygous genotypes TC and AG of the *ESR1* PvuII and XbaI polymorphisms were significantly elevated in T2DM patients compared with control individuals ($P = 0.04$, $P = 0.01$, respectively), while, the frequencies of C and G alleles of the *ESR1* PvuII and XbaI polymorphisms, respectively, were comparable among cases and controls ($P > 0.05$) (Table 2). However, upon Bonferroni's adjustment, only the heterozygous genotype AG of the *ESR1* XbaI remained significantly higher in diabetic group ($P = 0.01$).

## PvuII and XbaI Polymorphisms in *ESR1* gene and risk to type 2 diabetes

To estimate the effect of the genotypes on the disease, logistic regression analysis was performed using five genetic models adjusted to age and BMI: additive, dominant, codominant, recessive, and over-dominant. In the case of the PvuII T/C rs2234693 polymorphism, T2DM patients showed significantly different genotypes distribution compared to control group in three models, carriers of TC genotype had higher risk to T2DM compared to those of TT genotype (OR = 2.84, CI [1.32–6.11]; $P = 0.024$)

**Table 2 Allele frequencies and genotypes distribution of *ESR1* gene PvuII and XbaI polymorphisms in control and T2DM patients.**

| Allele/genotype | All subjects (%) | Control n (%) | T2DM n (%) | P-value |
|---|---|---|---|---|
| PvuII | | | | |
| T | 244 (57) | 133 (59) | 111 (54) | 0.46 |
| C | 184 (43) | 91 (41) | 93 (46) | 0.46 |
| TT | 70 (33) | 43 (38) | 27 (26) | 0.06 |
| TC | 104 (49) | 47 (42) | 57 (56) | 0.04 |
| CC | 40 (19) | 22 (20) | 18 (18) | 0.7 |
| XbaI | | | | |
| A | 247 (58) | 132 (59) | 115 (56) | 0.6 |
| G | 118 (42) | 92 (41) | 89 (44) | 0.7 |
| AA | 73 (34) | 44 (39) | 29 (28) | 0.09 |
| AG | 101 (47) | 44 (39) | 57 (56) | **0.01** |
| GG | 40 (19) | 24 (21) | 16 (16) | 0.35 |

Notes:
Bold numbers showed the significant correlation.
*P*-values lower than the Bonferroni threshold (*P* = 0.025) were considered statistically significant.

**Table 3 Association of *ESR1* PvuI and XbaI variants with T2DM.**

| Model | Genotype | OR (95% CI) | *P-value |
|---|---|---|---|
| PvuII SNP | | | |
| Codominant | T/T | 1 | |
| | T/C | 2.84 [1.32–6.11] | **0.024** |
| | C/C | 1.77 [0.66–4.73] | |
| Dominant | T/T | 1 | |
| | T/C–C/C | 2.50 [1.21–5.14] | **0.011** |
| Recessive | T/T–T/C | 1 | |
| | C/C | 0.96 [0.41–2.25] | 0.92 |
| Overdominant | T/T–C/C | 1 | |
| | T/C | 2.32 [1.18–4.55] | **0.013** |
| Log-additive | – | 1.48 [0.92–2.37] | 0.1 |
| Xba SNP | | | |
| Codominant | A/A | 1 | |
| | A/G | 2.15 [1.02–4.51] | 0.1 |
| | G/G | 1.18 [0.45–3.10] | |
| Dominant | A/A | 1 | |
| | A/G–G/G | 1.82 [0.91–3.66] | 0.089 |
| Recessive | A/A–A/G | 1 | |
| | G/G | 0.76 [0.32–1.80] | 0.53 |
| Overdominant | A/A–G/G | 1 | |
| | A/G | 2.03 [1.05–3.95] | 0.035 |
| Log-additive | – | 1.21 [0.76–1.92] | 0.42 |

Note:
* *P*-values were from logistic regression models adjusted for age and BMI, *P* < 0.025 was considered significant, bold numbers showed the significant correlation.

**Table 4 Demographic characteristics and biochemical measurements based on ESR1 PvuII and XbaI genotypes in diabetic women.**

| Parameter | PvuII genotypes | | | | XbaI genotypes | | | |
|---|---|---|---|---|---|---|---|---|
| | CC | TC | TT | P-value | AA | AG | GG | P-value |
| BMI (Kg/m$^2$) | 34.5 (1.8) | 33.7 (0.69) | 34.1 (1.3) | 0.91 | 34.5 (1.24) | 33.5 (0.69) | 34.8 (1.9) | 0.67 |
| SBP (mmHg) | 143.5 (3.50) | 135.5 (2.2) | 133 (2.9) | 0.09 | 134.6 (2.75) | 135.3 (2.27) | 142.4 (3.8) | 0.25 |
| DBP (mmHg) | 79 (3.0) | 78.8 (1.4) | 78.7 (1.6) | 0.93 | 79.9 (1.65) | 78.5 (1.4) | 78.1 (3.1) | 0.79 |
| FBS (mg/dl) | 169.85 (13.2) | 162.82 (6.84) | 154.44 (8.1) | 0.59 | 159.24 (9.39) | 159.14 (6.36) | 176.21 (13.98) | 0.46 |
| HbA1C | 8.28 (0.47) | 7.86 (0.26) | 7.72 (0.38) | 0.62 | 7.83 (0.4) | 7.78 (0.24) | 8.46 (0.52) | 0.46 |
| TC (mg/dl) | 202.42 (11.68) | 188.29 (5.3) | 192.33 (7.39) | 0.45 | 191.52 (6.84) | 189.57 (5.59) | 200.6 (11.78) | 0.65 |
| TG (mg/dl) | 250.07 (62.75) | 192.87 (19.25) | 164.45 (6.84) | 0.2 | 164.9 (6.37) | 212.01 (27.22) | 191.75 (15.81) | 0.42 |
| LDL (mg/dl) | 118.3 (7.69) | 105.95 (3.29) | 105.59 (4.78) | 0.19 | 106.1 (4.47) | 106.57 (3.53) | 116.78 (7.48) | 0.36 |
| HDL (mg/dl) | 38.56 (2.8) | 41.58 (1.19) | 44.55 (1.62) | 0.11 | 44.38 (1.59) | 41.14 (1.2) | 39.69 (3.04) | 0.2 |

Notes:
TC, total cholesterol; TG, triglyceride; HDL-C, high-density lipoprotein cholesterol; LDL-C, low-density lipoprotein cholesterol.
Values are presented as mean (standard error); P-value was obtained by ANOVA, $P < 0.05$ was considered significant.

**Table 5 Estimated haplotype distributions in cases (T2DM) and control groups.**

| PvuII | XbaI | Frequency (case) | Frequency (control) | Frequency (total) | OR (95% CI) | P-value | Global P |
|---|---|---|---|---|---|---|---|
| T | A | 0.5391 | 0.557 | 0.5485 | 1 | – | 0.14 |
| C | G | 0.4312 | 0.374 | 0.4013 | 1.36 [0.84–2.21] | 0.21 | |
| C | A | 0.0247 | 0.0323 | 0.0286 | 2.28 [0.62–8.37] | 0.22 | |
| T | G | 0.005 | 0.0367 | 0.0216 | 0.21 [0.02–2.66] | 0.23 | |

(Table 3). In the dominant model, the TC+CC genotype carriers was significantly at higher risk to T2DM than TT genotypes (OR = 2.50, CI [1.21–5.14]; $P = 0·011$). In the over-dominant model (TT+CC vs. TC), the association was also significant (OR = 2.32, CI [1.18–4.55]; $P = 0.013$) (Table 3). The analysis of the Xba rs9340799 A/G polymorphism showed a trend for association with T2DM in the over dominant model (AA+GG vs. AG) (OR = 2.03, CI [1.05–3.95]; $P = 0.035$) (Table 3).

Biochemical characteristics of T2DM individuals according to *ESR1* PvuII and XbaI genotypes are shown in Table 4. It demonstrated insignificant association of the investigated parameters across genotypes of both polymorphisms ($P > 0.05$). A stratified analysis of combined genotypes with respect to different genetic models also revealed no significant association with any biochemical or clinical parameter (data not shown).

## LD estimation between *ESR1* SNPs and haplotype analysis

The polymorphisms tested were in LD (normalized Lewontin's D' = 0.91). Among all subjects, the most common haplotype, TA, had a frequency of 55%, and the CG haplotype had a frequency of 40% while two haplotypes CA and TG had frequencies of 2.9% and 2.2%, respectively. The frequencies of the four haplotypes were not significantly different in the T2DM patients compared with control group as shown in Table 5. In logistic

**Table 6 Association of *ESR1* PvuII and XbaI variants with CVD among T2DM cases.**

| PvuII SNP | | CVD | | | |
|---|---|---|---|---|---|
| **Model** | **Genotype** | **Yes (%)** | **No (%)** | **OR (95% CI)** | ***P*-value** |
| Dominant | T/T | 54.5 | 23.1 | 1 | |
| | T/C–C/C | 45.5 | 76.9 | 0.25 [0.07–0.93] | 0.039 |
| Recessive | T/T–T/C | 90.9 | 81.3 | 1 | |
| | C/C | 9.1 | 18.7 | 0.41 [0.05–3.57] | 0.36 |
| Overdominant | T/T–C/C | 63.6 | 41.8 | 1 | |
| | T/C | 36.4 | 58.2 | 0.42 [0.11–1.57] | 0.19 |
| **Xba SNP** | | **CVD** | | | |
| **Model** | **Genotype** | **Yes (%)** | **No (%)** | **OR (95% CI)** | ***P*-value** |
| Dominant | A/A | 54.5 | 25.3 | 1 | |
| | A/G–G/G | 45.5 | 74.7 | 0.26 [0.07–0.99] | 0.049 |
| Recessive | A/A–A/G | 100 | 82.4 | 1 | |
| | G/G | 0 | 17.6 | 0.00 [0.00–NA] | 0.036 |
| Overdominant | A/A–G/G | 54.5 | 42.9 | 1 | |
| | A/G | 45.5 | 57.1 | 0.64 [0.18–2.31] | 0.49 |
| **PvuII/XbaI** | | **CVD** | | | |
| | **Haplotype** | **Yes (%)** | **No (%)** | **OR (95% CI)** | ***P*-value** |
| T/A | 73 | 52 | 1 | | |
| | C/G | 23 | 46 | 0.28 [0.09–0.90] | 0.035 |
| | C/A | 4 | 2 | 2.75 [0.22–33.96] | 0.43 |

**Note:**
NA, not applicable.

regression analyses, adjusting for the same potential confounders (age and BMI) used in the genotype models, none of the four possible haplotypes were associated with increased risk of diabetes (global $P = 0.14$) (Table 5).

## Association of *ESR1* PvuII and XbaI Polymorphisms with CVD in diabetic group

The association of PvuII and XbaI SNPs within the *ESR1* gene and CVD risk was evaluated among diabetic group ($n = 102$) using logestic regression after adjustment of age and BMI. Under the dominant model, the combined PvuII TC+CC genotypes have a trend to lower risk of CVD compared with TT genotype (OR = 0.25, CI [0.07–0.93]; $P = 0.039$). A similar trend was observed in the combined genotypes (AG+GG) of the *ESR1* XbaI compared to AA (OR = 0.26, CI [0.07–0.99]; $P = 0.049$) as shown in Table 6. The frequency of CG haplotype was statistically higher in diabetic patients without CVD (46%) compared to those with CVD (23%) (OR = 0.28, CI [0.09–0.90]; adjusted $P = 0.035$). On the other hand, we did not find any association of *ESR1* PvuII and XbaI polymorphisms with the other T2DM complications including nephropathy, retinopathy, and diabetic foot (data not shown).

## DISCUSSION

The prevalence of diabetes has increased worldwide, most markedly in the world's middle-income countries which reflects an increase in associated risk factors such as being overweight or obese (*World Health Organization (WHO), 2016*). At menopause, women experience health challenges due to estrogen deficiency. The risk of T2DM is increasing with post-menopause status interacting with other risk factors, that is, hypertension, dyslipidemia, and obesity (*Ren et al., 2018*). Estrogen may regulate insulin action directly via actions on insulin-sensitive tissues. In skeletal muscle, ESR1 is thought to have a positive effect on insulin signaling and GLUT4 expression. It has been shown that stimulation of estrogen receptor with its agonist propylpyrazoletriyl increased insulin-stimulated glucose uptake in skeletal muscles (*Gorres et al., 2011*). Although not all studies are in agreement, PvuII and XbaI and other polymorphisms across the *ESR1* gene have been associated with risk of T2DM as reported in the Chinese (*Huang et al., 2006*), African–Americans, European–Americans (*Sale et al., 2004*), Hungarians (*Speer et al., 2001*), and Egyptian women (*Motawi et al., 2015*). In this study, the frequency for *ESR1* PvuII and XbaI alleles and genotypes in Palestinian women ($n = 214$) was evaluated. Our results reported that the *ESR1* PvuII C and XbaI G allele frequencies were similar to other studies among Arab population (*El-Beshbishy et al., 2015*; *Motawi et al., 2015*). However, several studies showed distinct prevalence for these two polymorphisms, particularly the XbaI alleles (*Ganasyam et al., 2012*). Discrepancies in allelic frequencies between reports may be attributed to the ethnic variability of the studied populations, heterogeneity of the analyzed diseases and sample size.

Most often, deviation from the Hardy–Weinberg equilibrium (HWE) is caused by a small sample size and poor genotyping quality. In this study, minimization of genotyping errors was achieved by amplicon based NGS, It provides a lower cost per sample alternative to restriction fragment length polymorphism (RFLP), probe based Taq-man PCR and Sanger sequencing. RFLP, a traditional genotyping method, requires a high quality, and quantity of DNA sample and numerous steps including amplification, digestion with restriction enzymes and gel electrophoresis which can be a laborious and time-consuming process. Moreover, confirmatory sequencing must be performed when a sample shows confused banding pattern particularly the bands of low molecular size. In our analysis, DNA sequences with a quality score of more than 20 (>Q20), Q20 represents an error rate of 1 in 100 with a corresponding call accuracy of 99%, were selected. Amplicon based NGS requires advanced instrumentation, however and despite this limitation, we believe that it is the method of choice for genotyping studies and for rapid genetic screening of several hundred to thousands of samples.

To our knowledge, this is the first report to investigate whether *ESR1* PvuII and XbaI alleles and their genotypes and haplotypes are associated with T2DM among Palestinian women.

In this study, the additive, dominant, co dominant, and recessive genotypic models of association were tested, all were adjusted for age and BMI. The *ESR1* PvuII variant, supported evidence for association with risk of T2DM in three genotypic models

($P < 0.025$) but the strongest association was observed in the overdominant model, in which heterozygous carries confer a higher risk compared to homozygotes (Table 3). Although such association toward the heterozygotes could result from genotyping error, this is unlikely to be the case in our study as we used variant specific probes as described above.

One possible explanation is that the variant allele may have a very strong dominant effect so that there is little difference between the effects of the variant homozygotes and heterozygotes. Given that the majority of participants in both cases and controls were heterozygotes, the real effect among the variant homozygotes can only be assessed by much larger study in the future. On the other hand, the over dominant model (upon Bonferroni correction) showed a trend for association of XbaI SNP with T2DM (OR = 2.03, CI [1.05–3.95], $P = 0.035$) which was unexpected given the tight LD between these two polymorphisms. These results could also be attributed to the poor statistical power due to small sample size. Considering the associations of both SNPs found in other studies, we believe that further study involving a larger number of participants is needed to elucidate the exact role of these polymorphisms in susceptibility to T2DM. However, in 2018, a meta-analysis including eight studies indicated that PvuII, rather than XbaI polymorphism, was associated with T2DM (OR = 0.673, 95% CI [0.550–0.823]). In that study, the C allele of PvuII polymorphism showed a protective role in T2DM in Chinese people while the G allele of XbaI polymorphism is related to a reduced risk for T2DM in Caucasian population (*Yang et al., 2018*). Thus, we believe that physiological pathway to diabetes may vary among different population and those differences are reflected in part by genetic differences. Therefore, the results of our study should be considered exploratory and confirmed by additional studies including other polymorphisms/genes as one single gene polymorphisms are not enough to answer the pathogenesis of a complex disease such as T2DM. Further, it is well known that estrogens regulate the cardiovascular system via their direct effects on the vessel wall and indirect effects on total cholesterol and triglycerides metabolism (*Liping et al., 2013*). Among diabetic group, stratified analyses based on *ESR1* PvuII and XbaI genotypes revealed no evidence for association with lipid levels (TC, TG, HDL, LDL). Moreover, we did not find any association between the two polymorphisms (PvuII and XbaI) and TC in control group ($n = 114$). One limitation of this study is the lack of information about subjects who received hormone replacement therapy and/or lipid lowering agent which may affect their serum lipid profile, therefore, we were unable to study the interaction between these variants and hormone replacement therapy on lipid profile which was, however, beyond the scope of this study. It is reported that the effect of PvuII and XbaI polymorphisms on serum lipids depend on several other genes not on *ESR1* gene alone. Similarly, *Matsubara et al. (1997)* and *Almeida et al. (2006)* did not find any association between the two polymorphisms and serum lipid profile in post-menopausal women (*Almeida et al., 2006*; *Matsubara et al., 1997*). A recent study conducted on a cohort of post-menopausal Brazilian women reported no effects of PvuII SNP of *ESR1* gene on patient's serum TC, LDL, HDL, and TG, while XbaI SNP was associated to changes in TG and total lipids mainly in obese and overweight woman (*Gomes-Rochette et al., 2017*). In contrast,

Egyptian study showed that both PvuII and XbaI SNPs have been associated with increased levels of triglycerides, total cholesterol, and LDL (*Motawi et al., 2015*).

The mechanism by which intronic polymorphisms of *ESR1* gene might confer increased risk of T2DM is not fully understood. However, it is reported that the PvuII and XbaI polymorphisms change the *ESR1* gene expression by altering the binding of its own transcription factors. Moreover, LD between these two polymorphisms with other polymorphisms in the *ESR1* gene, such as TA tandem polymorphism in the promoter region could affect gene expression or function (*Herrington et al., 2002*). Polymorphisms in *ESR1* were first thought to be potential risk factors for the development of CVD, but a meta-analysis including 10 case-control studies demonstrated that the *ESR1* SNPs PvuII and XbaI are not associated with risk of CVD (*Morselli et al., 2017*). In this study we have noted an inverse trend between these polymorphisms and risk of CVD in diabetic group. Our findings cannot be a definitive conclusion to identify *ESR1* genotype/haplotype that could be useful in identification of diabetic women that are more prone to develop CVD due to small sample size with limited statistical power to detect interactions or perform subgroups analyses which was the most obvious limitation of this study.

## CONCLUSIONS

The present study suggests that the PvuII polymorphism in the *ESR1* gene is associated with increased risk of type 2 diabetes but not with lipid profile. The interaction between XbaI genotype and T2DM still need to be clarified. Identification of genetic risk markers can be used in defining high risk subjects and in prevention trials. The mechanism by which the *ESR1* PvuII and XbaI polymorphisms might be related to CVD-among diabetic group-needs further investigation with larger sample size. Also, future studies are warranted to replicate our results and to explain the influence of these intronic polymorphisms on diabetes susceptibility in Palestine.

## ACKNOWLEDGEMENTS

The authors gratefully acknowledge the UNRWA outpatient clinic members who helped to the patients' recruitment, and all study participants.

### Funding
The authors received no funding for this work.

### Competing Interests
The authors declare that they have no competing interests.

### Author Contributions
- Suheir Ereqat conceived and designed the experiments, analyzed the data, contributed reagents/materials/analysis tools, prepared figures and/or tables, authored or reviewed drafts of the paper, approved the final draft.

- Stéphane Cauchi conceived and designed the experiments, analyzed the data, authored or reviewed drafts of the paper, approved the final draft.
- Khaled Eweidat performed the experiments, approved the final draft, collect the blood samples and extract the DNA.
- Muawiyah Elqadi performed the experiments, approved the final draft, collect the data from patients medical records.
- Abedelmajeed Nasereddin conceived and designed the experiments, contributed reagents/materials/analysis tools, prepared figures and/or tables, authored or reviewed drafts of the paper, approved the final draft.

## Human Ethics

The following information was supplied relating to ethical approvals (i.e., approving body and any reference numbers):

The study protocol was approved by Al-Quds University Research Ethics Committee (Rf no. 2/SRC/4).

## Data Availability

Anonymized raw data are available in a Supplemental File.

## Supplemental Information

Supplemental information for this article can be found online at http://dx.doi.org/10.7717/peerj.7164#supplemental-information.

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
