# Peer review of "Estrogen receptor 1 gene polymorphisms (PvuII and XbaI) are associated with type 2 diabetes in Palestinian women"

_PeerJ, doi:10.7717/peerj.7164_

## Round 0.1 · original submission · Major Revisions

Please heed all of the reviewer's comments, especially the concerns of reviewer #2, when submitting a new version of the manuscript.

Reviewer 1 ·

Basic reporting

The manuscript entitled “Estrogen receptor 1 gene polymorphisms (PvuII and XbaI) are associated with type 2 diabetes in Palestinian women” by Ereqat et al describe the association of ESR1 gene polymorphisms with T2DM in Palestinian women. In this manuscript, the atuhors conducted a cross sectional study of 214 Palestinian women (102 with T2DM and 112 non diabetic controls), aged 45–70 years, characterised for blood glucose, blood lipid, blood pressure and BMI, and genotyped for two ESR1 gene polymorphisms, the PvuII (rs2234693) and XbaI (rs9340799). The manuscript is generally well written, the protocol appears to be appropriate and well-conducted, and the statistical analysis were performed appropriately. However, the following issues need to be addressed and clarified before the manuscript could be considered for publication:

1. The authors highlighted the importance of estrogen and estrogen receptors, as shown in the manuscript (line 59-65, first paragraph of Introduction): “Recently, several reports have revealed the role of estradiol in regulating energy metabolism and highlighting the role of the estrogen receptors which are distributed in many tissues, including the pancreatic islets, skeletal muscle, adipose tissue, and cardiovascular system (Hevener et al. 2018). It was evident that estrogens exert certain physiological and metabolic disorders including insulin resistance, dyslipidemia, and hypertension (Howard et al. 2003; Lindsay & Howard 2004), which all participate in the pathogenesis of T2DM.” However, the authors did not provide enough information on how and by which mechanism that estrogen and its receptors influenced metabolic disorder. For the mechanism, please see Meyer et al, 2011, Acta Physiol (Oxf), doi:10.1111/j.1748-1716.2010.02237.x, for review.

2. Why did the authors select the two ESR1 SNPs, PvuII (rs2234693) and XbaI (rs9340799) for their study? What is the importance of studying these variants? Please described in Introduction.

Experimental design

1. The description of the study population is not clear. The authors should explain how the participants of this study were enrolled in detail. Are there inclusion and exclusion criteria?

2. Methods of BMI, SBP, DBP, and FBS measurements were not described in the text. The authors should describe those measurements clearly in Methods.

3. The authors should describe the HWE calculation in Methods.

4. The authors used linear regression for multivariate analysis, as described in Methods, Statistical analysis section, line 134-135: “Multivariate linear regression models taking into account age and BMI were performed to assess associations between SNPs and T2DM complications”. Are the parameters distributed normally?

5. In Methods, Statistical analysis section, line 135-136, the authors stated at the bottom of this section that "Comparison of the tested SNPs was statistically significant at P < 0·05." However, the authors tested 2 different SNPs of ESR1 gene. To minimize the chances of obtaining type I error, the authors should perform Bonferroni correction, such that the statistical significance level shall be 0.05/2=0.025.

6. The sample size was rather small. Therefore, please provide statistic power of your sample. "GeneticsDesign" package (Purcell et al, 2003, Bioinforma Oxf Engl, 19:149–150) could be use for power calculation

Validity of the findings

1. In Result (line 140-141), the authors stated that “...all clinical parameters showed statistically significant increase in T2DM patients (P<0.05)”. From Table 1 one can see that the control group is 11 years younger than the T2DM group. Will the control group developed T2DM eventually? Is the significant differences of SBP and TC between control and T2DM groups related to T2DM or age?

2. The authors did not present any Table related to the “Association of ESR1 PuVII and Xba Polymorphisms with T2DM complications” section (Result, line 177-186). The authors should provide a Table that referred to the result described in this section. The authors should also described what are the CVD risk and T2DM complications that they mention in this section.

Additional comments

1. Line 59 is missing.
2. Line 84: “A meta-analysis study investigated the new-onset T2DM in postmenopausal women following estrogen replacement therapy,...”, should be “A meta-analysis study investigating the new-onset of T2DM in postmenopausal women following estrogen replacement therapy,...”
3. Line 96-98: Please provide a reference for this statement “Diagnosis of T2DM was based on World Health Organization (WHO) criteria (fasting plasma glucose126 mg/dl and/or currently on treatment for diabetes)”.
4. Line 103-104: “Genomic DNA was extracted from whole blood (300 μl) using genomic DNA purification kit QIAamp...”, should be “Genomic DNA was extracted from whole blood (300 μl) using genomic QIAamp DNA purification kit...”.
5. Line 108-109: “Both primers were modified with over hanged Illumina adaptor sequences at the 5′ ends (italic bolded), should be “Both primers were modified with over hanged Illumina adaptor sequences at the 5′ ends (italic bolded, Table S1)”.
6. Please use consistently “Control group” instead of: “healthy control” (line 34), or “healthy subjects” (line 40, 156, 174) , or “healthy women” (line 233).
7. Line 139: Table 1, with uppercase T.
8. Line 147: P>0.05, with uppercase P.
9. Line 152: typographical error: “PuVII and Xba” should be “PvuII and XbaI”.
10. Line 158: Table 3, not 2.
11. Line 165: Table 4, with uppercase T and without brackets.
12. Line 174: Table 5, with uppercase T and without brackets.
13. Line 189: obese, with lowercase o.
14. Line 201-202: allele frequencies.
15. Line 204: XbaI.

Reviewer 2 ·

Basic reporting

The submitted manuscript aimed at investigating the possible role played by two ESR1 gene polymorphisms in conferring Palestinian women susceptible for T2DM. the study population comprised 104 patients previously diagnosed with T2DM, and 112 control healthy females.
The results of the manuscript are interesting. However, extensive revisions are needed in order to make it publishable.

Experimental design

The study is a case control study, although not stated in the methods section.
Selection criteria for the cases and controls is note stated. A number of pieces of information needs to be introduced, including the type of study, sampling method, inclusion criteria, and exclusion criteria.
The cases and controls were not matched in terms of age, which is a major risk factor for T2DM??
Were the cases recruited from one UNORWA clinic? Which one?
Were all cases recruited from the same governorate/city??
How were controls recruited? Are they healthy only in terms of T2DM?
Did the authors confirm the diagnosis of cases? If yes, then how?
Did the authors confirm the controls are not diabetic??
What data was collected from the patients and/or their records in the primary health care clinic? How?
Did the authors take the permission of UNORWA to recruit the cases from its clinics?

The commercial names and manufacturers should be stated for all kits used in the experiments (example line 103, 112, 118).

What methods were used for determining the FBG, CL, LDL, HDL, TG levels??

The statistical analysis looks sufficient.

Validity of the findings

In order to be valid, the authors should discuss their findings as follows:

An important requirement for studying allele frequencies in population genetics is a large sample size. Therefore, the authors should discuss how their results would be valid in light of their sample size before drawing important conclusions.
Over-dominance (or heterozygous advantage) is a way by which allele frequencies become different from those expected by Hardy Weinberg equations However, the authors report that the frequencies are in HW equilibrium (line 147). Moreover, how do the authors explain their predicted model? In other words, how can polymorphic ESR1 alleles give an advantage for heterozygotes over wildtype and mutant homozygotes as predicted by your results of overdominance?

Additional comments

Intensive English language revisions should be carried out.
In the abstract section, the background and methods should be rewritten in order to reflect the justification of the study and clarify the methods used in the study.
The Introduction section needs to be rewritten in a more focused and concise way. The sentences are some times very long and not clear. For example, in line 78-82 the sentence “A recent study conducted on three refugee camps in the West Bank, revealed that the overall prevalence of obesity and overweight was 63.1% while the prevalence of MetS among obese and overweight was 69.4% with no significant differences between men and women (Damiri et al. 2018) indicating that they are at high risk of both cardiovascular diseases and type 2 diabetes”

The references are sometimes old.

---

## Round 0.2 · Minor Revisions

Please made amendments as suggested by the reviewer, in order to have the paper accepted for publication.

Reviewer 1 ·

Basic reporting

No comment

Experimental design

No comment

Validity of the findings

No comment

Additional comments

Some part of the revised Introduction section is too long and needs to be shortened in a more focused and concise way (line 60-73).

---

## Round 0.3 · accepted · Accept

Your article has now been deemed acceptable for publication in PeerJ.

# Reviewer 1 ·

Basic reporting

No comment

Experimental design

No comment

Validity of the findings

No comment

Additional comments

The revised Introduction is acceptable.